# Maternal adverse childhood experiences and perinatal outcomes: A retrospective inceptive cohort study

Amanda Troëng[1], Jessica Dolk[1], Marie-Therese Vinnars[2], Johan Hallqvist[1], Per Kristiansson[1]*

**1** Department of Public Health and Caring Sciences, Uppsala University, Uppsala, Sweden, **2** Department of Clinical Sciences, Umeå University, Umeå, Sweden

* per.kristiansson@uu.se

## Abstract

### Introduction

Adverse childhood experiences (ACEs) are associated with poor health and social outcomes in adulthood. However, research on the relationship between ACEs and perinatal outcomes remains limited, and the effect of cumulative ACEs on perinatal outcomes has not yet been established. This study examines the association between maternal exposure to multiple ACEs and perinatal outcomes.

### Material and methods

The study comprised a cohort study of 1,253 women enrolled in the national prenatal healthcare program in Sweden 2012-2013. In early pregnancy and one year after childbirth the women completed a total of 3 questionnaires that included questions on ACEs, mode of delivery, and birth complications. ACEs were grouped as involving 0, 1-3, or ≥4 categories according to previously defined measures. Multiple ordinal logistic regression analyses were used to compare perinatal outcomes across ACE categories adjusted for a minimal sufficient set of confounders as well as sensitivity analyses.

### Results

The women had an average age of 32, and the majority were multiparous, of Swedish origin, highly educated, non-smokers and in a relationship. The proportions of women with adverse childhood experiences in the 0, 1-3, and ≥4 categories were 42%, 46% and 12%, respectively. On the whole, the adjusted odds ratios (aOR) were highest for women exposed to adverse childhood experience in ≥4 categories, with women exposed to 0 as reference, and with aORs for women exposed to 1-3 categories in between. In women with ≥4 ACEs categories, the strongest associations were

**Data availability statement:** All relevant data are within the paper and its Supporting information files.

**Funding:** The author(s) received no specific funding for this work.

**Competing interests:** The authors have declared that no competing interests exist.

**Abbreviations:** ACEs, adverse childhood experiences; Q, questionnaire.

found for the outcomes emergency Caesarean delivery (aOR 2.02, CI 1.13-3.63), bleeding >1000 ml in connection with Caesarean delivery (aOR 3.54, CI 1.01-12.39), preeclampsia (aOR 4.21, CI 1.73-10.25) and requiring antibiotics (aOR 3.14, CI 1.19-8.32).

## Conclusions

Multiple ACEs were associated with higher rates of adverse perinatal outcomes. The maternal health services need to identify these individuals and provide extra care to mitigate their risks.

## Introduction

Adverse childhood experiences (ACEs) encompass conditions and circumstances that contributes to toxic life stress, potentially causing lasting detrimental effects on health that persist through adulthood [1–3]. Exposure to ACEs is associated with an increased risk a number of health issues, including public health diseases such as stroke and cancer [1,4,5], increased risk behaviors [6], and lower socioeconomic outcomes [7]. The impact of ACEs is cumulative, with each additional experience increasing the likelihood of adverse health effects. Approximately 60% of the population reports experiencing at least one ACE, while 10% report four or more [2]. Further, ACEs often occur together, rather than in isolation [8].

Pregnancy is a vulnerable time, marked by increased sensitivity to both positive and negative factors [9]. Recently, growing attention has been pain to the adverse effects of ACEs on pregnancy and perinatal outcomes. Toxic stress in early life can induce alterations in the nervous, endocrine and immune systems [10], as well as affect biological markers associated with life stress and aging [11]. Epigenetic modifications and allostatic load have both been proposed as underlying biological mechanisms [10,12].

ACEs have previously been linked to increased risky health behaviors, unintended pregnancies, poor prenatal health, prematurity, lower birth weight, and hypertensive disorders [13–19]. However, so far research on the impact of multiple ACEs on perinatal outcomes remains limited.

The aim of this study was to examine the association between the extent of self-reported maternal ACEs and perinatal outcomes.

## Materials and methods

This cohort study is based on questionnaires completed by 1,253 pregnant women participating in the Swedish national antenatal screening program 2012–2013. The study involved 153 of 196 invited antenatal clinics located in both rural and urban areas of central Sweden. During the recruitment period, from September 2012 to July 2013, 5,796 women were enrolled. Of these, 3,389 (62%) agreed to participate and met the inclusion criteria, which excluded those who did not speak Swedish, experienced miscarriage, or had pregnancies involving multiples.

Midwives at the participating antenatal clinics distributed the first questionnaire (Q1) to the participants. The second (Q2) and third (Q3) questionnaires were sent by post, together with a prepaid return envelope, to those who had responded to the previous questionnaire. Non-responders received a reminder within two weeks. All participants who completed Q1 (3,389) received Q2. Those who completed Q2 (2,018/3,389) were subsequently sent Q3, of which 1,253 women (1,253/2,018) completed the final questionnaire Q3. On average, Q1 was completed at 9.6 weeks of gestation, Q2 at 31.9 weeks of gestation and Q3 at 52.8 weeks postpartum.

The first questionnaire (Q1) gathered information on age, weight, height, ethnic origin, education level, and the number of previous pregnancies and deliveries. The second questionnaire (Q2) focused on Estimated Due Date, determined by routine ultrasound at 18 weeks' gestation. The third questionnaire (Q3) explored ACEs, details of the recent pregnancy and delivery, date of delivery, weight and length of the infant, alcohol consumption, smoking habits, relationship status, and household income.

Women who completed Q3 and were thus included in the study differed as a group compared to those who answered Q1 only. Among participants who completed both questionnaires Q1 and Q3, a greater proportion reported higher levels of education (51% vs 37%, p < 0.0001), European origin (18% vs 27%, p < 0.0001), cohabiting (99% vs 97%, p = 0.0009), weekly alcohol consumption (80% vs 72%, p < 0.0001), and non-smoking (14% vs 23%, p < 0.0001), as compared with participants completing only Q1. However, their body mass index (25.2 kg/m$^2$ vs 25.2 kg/m$^2$, p = 0.634) and the proportion of participants reporting the highest household income (31% vs 29%, p = 0.119) remained similar between the two groups.

## Exposure variable

The survey of ACEs originates from a study conducted by Felitti *et al.* 1998, including 10 ACE categories [1]. To create a Swedish version, the questions were translated from English to Swedish and then translated back to English to test for accuracy. This process involved three translators: one native English speaker with Swedish as a second language, and two native Swedish speakers with English as their second language. The translated questionnaire was then compared to the original to ensure accuracy.

All ACE questions referred to experiences during the first 18 years of life. Each ACE category contained between one and eight questions, coded as either experienced or not experienced. The number of experienced ACE categories was summed to calculate an ACE score, ranging from zero to 10.

To examine the impact of exposure to multiple ACE categories, a known major risk factor for numerous health conditions [2], the women were divided into one of three groups: those with 0 ACE categories, those with 1–3 ACE categories, and those with four or more ACE categories. Each ACE category was initially coded as not experienced unless any of the criteria, described in an earlier study by Andersson *et al* in 2021 [3] were met.

## Outcome variables

*Onset of labor* was assessed by the study-specific question "How did labor start?" Participants selected from the following options: spontaneous, induction or planned Caesarean section.

*End of labor* was assessed with the study-specific question "What type of delivery did you have?" Response options included vaginal delivery, vacuum extraction, forceps delivery, and emergency Caesarean section. Since none of the participants reported a forceps delivery, this option was omitted from data reporting.

*Total number of Caesarean sections* was calculated as the combined total of elective and emergency Caesarean sections.

*Obstetric complications* were captured through a multiple-choice question: "Did any other complications arise in connection with the delivery? (Check all options that apply)". Participants could select all applicable complications they experienced from the following options: bleeding >1000 ml post-partum hemorrhage, retained placenta, placental abruption,

preeclampsia, obstetric anal sphincter injury, infection requiring antibiotics, neonate needing respiratory support, admission to the neonatal intensive care unit (NICU), other and none. The option "other" provided no further details and was selected by 93 of 1,235 women (7.5%).

Information on the *fetal* weight (in grams), length (in centimeters), and congenital conditions was provided by the mothers. The corresponding questions were: "How much did your child weigh at birth", "How long was your child at birth", and "Did your child have any malformations, injuries, or illnesses at birth?",

The estimated due date was reported in Q2. The estimated date of conception was calculated as 280 days prior to the estimated due date. The actual date of birth was reported by mothers in Q3. Pregnancy duration in days was determined by calculating the time between the actual date of birth and the estimated date of conception.

### Co-variates

Maternal body mass index (BMI) was calculated as the weight (in kilograms) divided by the height squared (in meters). Weight data was obtained from Q3, through the question, "What is your current weight? (in kilograms)". Height data was collected from Q1 during early pregnancy, which asked: "What is your height? (in centimeters)".

Non-Swedish origin was identified if either the participant or at least one of their parents was born outside of Sweden. This information was gathered in Q1 using the question: "Specify your and your parents' country of birth" with the response options: Sweden, another Nordic country, another European country, or a non-European country.

The level of education attained was assessed with a seven-point Likert scale based on the question: "What is the highest level of education you have completed?" The scale ranged from "no formal education" to "university studies with a doctoral degree". High education level was defined as having completed three or more years of university studies.

Monthly household income was assessed on an 11-point Likert scale in response to the question: "What is you household's monthly income before tax?" The income categories ranged from 0 SEK (Swedish krona) to over 100,000 SEK, increasing in increments of 10,000 SEK. High monthly household income was defined as being at or above the 75th percentile.

The postpartum partner status was assessed with the question "Do you have any partner today?" offering the following response options: "Yes, the same as when I got pregnant", "Yes, but not the same as when I got pregnant" and "No".

Cigarette smoking was addressed with the question "Do you smoke at present?" The available responses were "No", "Yes, I smoke every day" or "Yes, but not every day".

Weekly alcohol consumption was addressed with the question "Do you drink alcohol at present?" The available responses were "No", "Yes, every week", "Yes, but not every week".

Variables potentially influencing both exposure and outcome variables (confounders) were identified based on available scientific literature. Their mutual relationships were visualized using a causal diagram, specifically a directed acyclic graph (DAG) (Fig 1), created with the software dagitty.net (version 3.0). The selection of confounders followed established principles, such as pretreatment criteria and causal diagram [20]. Applying the pretreatment criteria, parental ethnic origin and socioeconomic status were identified as variables that timely precede exposure to adverse childhood experiences (ACEs). Using a causal diagram and the "back-door path criterion," these two factors were further confirmed as key to mitigating confounding bias. The diagram also suggested that variables such as a new partner, stress during pregnancy, alcohol consumption, smoking habits, BMI, education level, age, parity, and pre-pregnancy hypertension act as mediators. Adjusting for these mediators could introduce bias. Given the absence of data on parental socioeconomic status, ethnic origin was selected as the confounder for adjustment in the regression analyses.

A sensitivity analysis was conducted that included an alternative causality model based on other assumptions about the complexity among potential confounders as part of a chain of confounders or mediators. In this model, adjustments were made for age, high level of education, smoking habits, alcohol habits, BMI and number of previous pregnancies as well as adjustments in the causal diagram model.

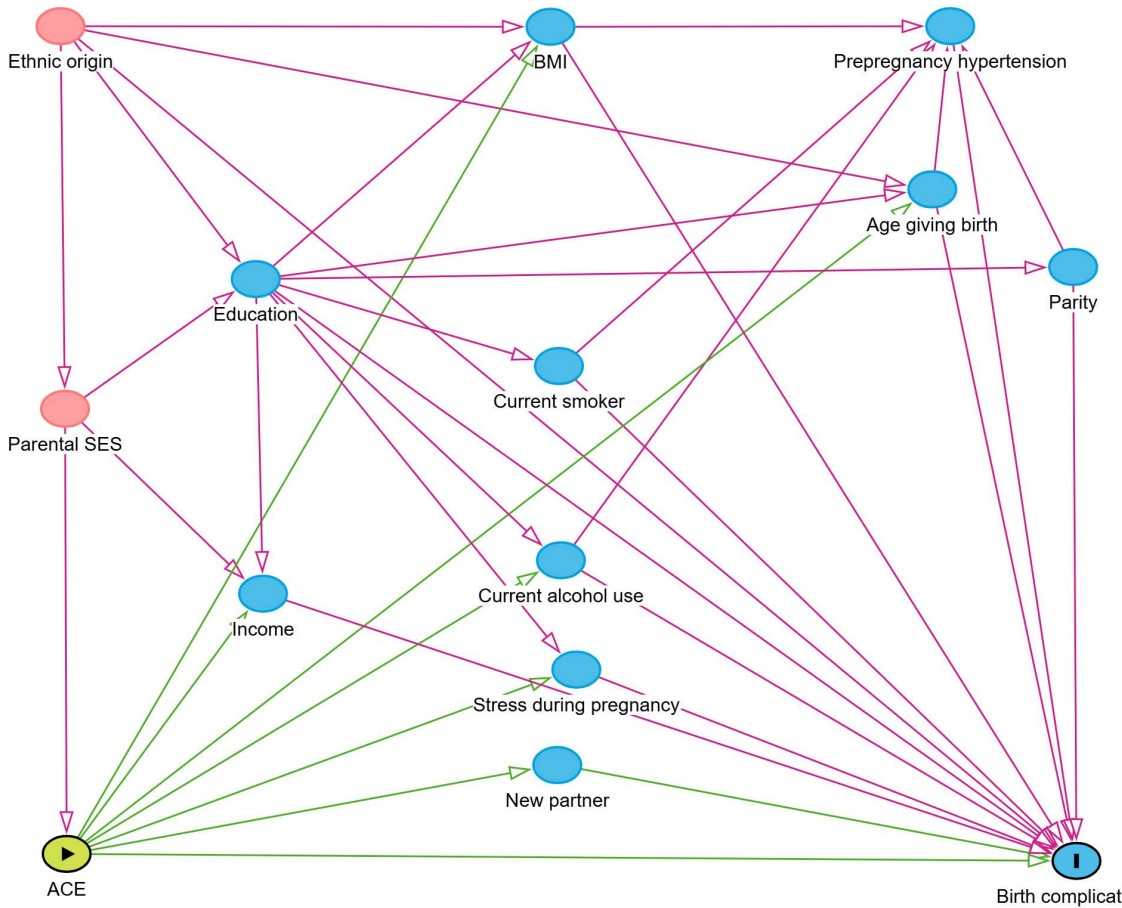

**Fig 1. Directed acyclic graph examining the causal relationship between adverse childhood experiences (ACEs) and perinatal outcomes.**
SES = socioeconomic status. Made using dagitty.net (version 3.0).

In addition, a sensitivity analysis was conducted to evaluate the robustness of the association to potential unmeasured and uncontrolled confounding, using the E-value calculation, by the formula E-value=*OR + sqrt (OR x (OR-1))* [21].

## Statistical analyses

Summary statistics were calculated by standard methods: frequencies for the categorical variables, median and interquartile range for ordinal and skewed data, and mean with standard deviation (SD) for continuous variables. To assess the probability of no difference between groups, $\chi^2$-test or Fisher's exact test was applied to categorical data, Mantel-Haenszel $\chi^2$-test for ordinal data and Students t-test for continuous data.

Multiple ordinal logistic regression analyses were used to evaluate the odds for several outcomes among mothers exposed to 1–3 ACE categories or ≥4 ACE categories, with women exposed to 0 ACE categories serving as the reference. The results were reported as crude odds ratios (OR) and adjusted odds ratios (aOR) along with their corresponding 95% confidence intervals (CI). Non-Swedish origin was the confounding factor adjusted for in the analyses. A p-value of <0.05 was considered statistically significant.

**Analysis of non-response.** To determine the influence of participants with missing data regarding any ACE category, a separate sensitivity analysis was performed in which all missing ACE categories were coded as "not experienced".

Statistical analyses were made using the SAS program, version 9.4.

### Ethics statement

The study received approval from the Regional Ethics Board in Uppsala, Sweden (Decision number 2010/085 and 2017/085/5). At antenatal clinics, midwives provided women with verbal and written information about the study. Those who agreed to participate were given the following consent form in Questionnaire 1: "I consent to participate in the study "How many children will I have when I grow up?" I have received written information about the study and have had sufficient time to consider my participation. I understand that I can withdraw from the study at any time. I consent to my personal data being processed as described in the patient information and acknowledge that my participation is entirely voluntary. I will receive a copy of the patient information. The return of completed questionnaires was regarded as informed consent.

### Results

Table 1 summarizes the reported exposure to adverse childhood experience (ACE) categories among the cohort of 1,253 mothers. The most common ACE category was parental separation or divorce, followed by physical abuse, emotional neglect, household mental illness, and substance abuse. The least common ACE category was having an incarcerated household member. Of the participants, 520 (41.5%) reported no ACE category, 577 (46.0%) reported 1–3 ACE

**Table 1. Categories of Adverse Childhood Experience (ACEs) reported by 1,253[1] women one year postpartum, with corresponding numbers and percentages provided.**

| ACE category | n (%) |
|---|---|
| Physical abuse | 275 (22.0) |
| Emotional abuse | 67 (5.4) |
| Sexual abuse | 109 (8.8) |
| Physical neglect | 73 (5.8) |
| Emotional neglect | 260 (20.8) |
| Witnessing domestic violence | 92 (7.4) |
| Household mental illness | 231 (18.5) |
| Household substance abuse | 232 (18.5) |
| Parental separation or divorce | 355 (28.4) |
| Incarcerated household member | 31 (2.6) |
| Accumulated ACE categories | |
| 0 | 520 (41.5) |
| 1 | 332 (26.5) |
| 2 | 160 (12.8) |
| 3 | 85 (6.8) |
| 4 | 67 (5.4) |
| 5 | 37 (3.0) |
| 6 | 20 (1.6) |
| 7 | 17 (1.4) |
| 8 | 11 (0.9) |
| 9 | 1 (0.1) |
| 10 | 3 (0.2) |

[1] In average data for each category was missing in four women (maximum eight).

categories, and 156 (12.5%) reported ≥4 ACE categories. Accordingly, the mothers were categorized into groups based on exposure to 0, 1–3 and ≥4 ACE categories.

Table 2 presents the characteristics of the mothers across the three ACE category groups. There is a progressive increase in BMI, non-Swedish origin, having a new partner after birth, and cigarette smoking across the groups. Conversely, there is a gradual decline in the proportions of mothers with the high education level, high monthly household income and weekly alcohol consumption. In average data for each category was missing in four women (maximum eight).

Table 3 outlines the reported outcomes for onset of labor, labor completion, and labor complications. Across the three ACE category groups, no significant differences were observed in onset of labor, although a trend toward less spontaneous onset of labor was noted (p = 0.061). Regarding end of labor, the proportion of vaginal non-instrument assisted births decreased gradually (p = 0.044), while emergency Caesarian deliveries increased across the ACE groups (p = 0.020). As for labor complications, the incidence of bleeding >1,000 ml (p = 0.016), placental abruption (p = 0.023), preeclampsia (p = 0.006), and requiring antibiotics (p = 0.014) progressively rose across the three ACE category groups. Notably, women with no ACE exposure reported no cases of placental abruption.

Table 4 presents the results of the multiple ordinal logistic regression analyses examining the causal associations between the exposure to 1−3 and ≥4 ACE categories and perinatal outcomes, using women with 0 ACE categories as the reference group. The adjusted odds ratios (aOR), controlled for origin, were highest for women exposed to ≥4 ACE categories, followed by those exposed to 1−3 ACE categories, compared to the reference group. The strongest associations were shown with the outcomes emergency Caesarean delivery (aOR=2.02, CI 1.13–3.63), bleeding >1000 ml (aOR=2.00, CI = 1.07–3.75), preeclampsia (aOR=4.21, CI = 1.73–10.25), and requiring antibiotics (aOR=3.14, CI 1.19-8-32). Further, for this group,

**Table 2. Characteristics of 1253 women one year after childbirth by number of Adverse Childhood Experiences (ACE) categories (women with 0, 1-3 and ≥4 categories of ACE).**

| Characteristic | All women | | Number of ACE categories | | | | | | p[2] |
|---|---|---|---|---|---|---|---|---|---|
| | n[1] | | n[1] | 0 | n[1] | 1-3 | n[1] | ≥4 | |
| Age (yr) | 1198 | 31.6 (4.8) | 497 | 31.8 (4.6) | 552 | 31.5 (4.9) | 149 | 30.9 (5.3) | 0.147 |
| BMI (kg/m[2]) | 1225 | 24.8 (4.9) | 508 | 24.4 (4.6) | 562 | 24.9 (4.9) | 155 | 25.7 (5.6) | *0.010* |
| Non-Swedish origin[3] | 1230 | 218 (17.7) | 512 | 75 (14.6) | 566 | 107 (18.9) | 152 | 36 (23.7) | *0.006* |
| High education level[4] | 1233 | 641 (52.0) | 514 | 303 (59.0) | 570 | 291 (51.0) | 149 | 47 (31.5) | *<0.001* |
| High household income[5] | 1226 | 409 (33.4) | 509 | 193 (37.9) | 562 | 181 (32.3) | 155 | 35 (22.6) | *<0.001* |
| In a relationship | 1243 | 1220 (98.2) | 519 | 512 (98.6) | 570 | 559 (98.1) | 154 | 149 (96.8) | 0.138 |
| Different partner[6] | 1220 | 11 (0.9) | 512 | 2 (0.39) | 559 | 4 (0.7) | 149 | 5 (3.4) | *0.005* |
| Cigarette smoking[7] | 1252 | 63 (5.0) | 519 | 15 (2.9) | 577 | 35 (6.1) | 156 | 13 (8.3) | *0.002* |
| Weekly alcohol use | 1252 | 880 (70.3) | 519 | 387 (74.6) | 577 | 406 (70.4) | 156 | 87 (55.8) | *<0.001* |
| Nulliparous | 1242 | 576 (46.4) | 517 | 233 (45.1) | 570 | 271 (47.5) | 155 | 72 (46.5) | 0.562 |

[1] Effective responses.

[2] The probability of no difference between the three groups, tested by Mantel-Haenszel Chi-Square statistic for categorical variables with two levels, non-parametric test for categorical variables with three levels and one-way analysis of variances for continuous variables.

[3] Defined as either the participant or at least one of the participant's parents being born outside of Sweden.

[4] Educational level, was reported on a 7 point Likert scale between no education and university studies with doctoral degree. High educational level was defined as ≥3 years of university studies and reported as median and interquartile range.

[5] Monthly household income, presented on an 11 point Likert scale where each point indicates 10000 Swedish Krona (SEK) between 0 SEK to ≥$10^5$ SEK/month. High household income was defined as being at or above the 75th percentile.

[6] As compared with the beginning of pregnancy.

[7] Coded as cigarette smoker if answered "yes, every day" or "yes, but not every day" to the question "do you currently smoke?"

Mean (standard deviation (SD)), number (%), and median (interquartile range (IQR)) are presented.

**Table 3. Proportions of perinatal outcomes reported by 1,253 women one year after childbirth. The women are grouped by exposure to 0, 1-3 and ≥4 Adverse Childhood Experiences (ACE) categories.**

| Outcome | All women | | ACE category exposure | | | | | | |
|---|---|---|---|---|---|---|---|---|---|
| | n[1] | | n[1] | 0 | n[1] | 1-3 | n[1] | ≥4 | p[2] |
| Small for gestational age | 1128 | 15 (1.3) | 468 | 5 (1.1) | 521 | 7 (1.3) | 139 | 3 (2.2) | 0.359 |
| Onset of labor | | | | | | | | | |
| Spontaneous | 1244 | 946 (76.0) | 514 | 405 (78.8) | 575 | 428 (74.4) | 155 | 113 (72.9) | 0.061 |
| Induction | 1244 | 200 (16.1) | 514 | 76 (14.8) | 575 | 93 (16.2) | 155 | 31 (20.0) | 0.146 |
| End of labor | | | | | | | | | |
| Vaginal birth, non-instrument ass.[3] | 1246 | 941 (75.5) | 517 | 405 (78.3) | 574 | 425 (74.0) | 155 | 111 (71.6) | *0.044* |
| Vacuum extraction | 1246 | 98 (7.9) | 517 | 42 (8.1) | 574 | 44 (7.7) | 155 | 12 (7.7) | 0.812 |
| Emergency Caesarean delivery | 1246 | 109 (8.8) | 517 | 37 (7.2) | 574 | 51 (8.9) | 155 | 21 (13.6) | *0.020* |
| Elective Caesarean delivery | 1244 | 98 (7.9) | 514 | 33 (6.4) | 575 | 54 (9.4) | 155 | 11 (7.1) | 0.327 |
| Complications of labor[4] | | | | | | | | | |
| No complication | 1231 | 865 (70.3) | 515 | 376 (73.0) | 563 | 393 (69.8) | 153 | 96 (62.8) | *0.018* |
| Bleeding >1000 ml | 1231 | 92 (7.4) | 515 | 30 (5.8) | 563 | 44 (7.8) | 153 | 18 (11.8) | *0.016* |
| Vaginal birth | 1023 | 63 (6.2) | 444 | 24 (5.4) | 458 | 28 (6.1) | 121 | 11 (9.1) | 0.183 |
| Caesarean delivery (all) | 204 | 27 (13.2) | 69 | 5 (7.2) | 104 | 15 (14.4) | 31 | 7 (22.6) | *0.032* |
| Retained placenta | 1231 | 51 (4.4) | 515 | 22 (4.3) | 563 | 18 (3.2) | 153 | 11 (7.2) | 0.398 |
| Placental abruption | 1231 | 6 (0.5) | 515 | 0 (0) | 563 | 4 (0.7) | 153 | 2 (1.3) | *0.023* |
| Preeclampsia | 1231 | 38 (3.1) | 515 | 11 (2.1) | 563 | 16 (2.8) | 153 | 11 (7.2) | *0.006* |
| Obstetric anal sphincter injury[5] | 1023 | 53 (5.2) | 444 | 25 (5.6) | 458 | 24 (5.2) | 121 | 4 (3.3) | 0.371 |
| Requiring antibiotics | 1231 | 43 (3.5) | 515 | 10 (1.9) | 563 | 25 (4.4) | 153 | 8 (5.2) | *0.014* |
| Child respiratory support | 1231 | 59 (4.8) | 515 | 18 (3.5) | 563 | 32 (5.7) | 153 | 9 (5.9) | 0.099 |
| NICU admission[5] | 1231 | 75 (6.1) | 515 | 26 (5.0) | 563 | 38 (6.8) | 153 | 11 (7.2) | 0.213 |
| Other | 1231 | 93 (7.6) | 515 | 34 (6.6) | 563 | 45 (8.0) | 153 | 14 (9.2) | 0.241 |

[1] Effective response.

[2] The probability of differences between the groups, tested by Mantel-Haenszel chi-square test.

[3] Vaginal birth, non-instrument assisted.

[4] In connection to delivery.

[4] In connection to vaginal delivery.

[5] Neonatal intensive care unit.

Number and proportion (%) are presented.

the adjusted odds ratio for having no labor complication was 0.62 (CI 0.42–0.91). These associations remained statistically significant with a similar effect estimation in a sensitivity analysis, except for requiring antibiotics, as presented in Table 5.

In a sensitivity multiple ordinal logistic regression analysis adjusting for age, high educational level, ethnic origin, smoking habits, alcohol habits, BMI and number of previous pregnancies the adjusted odds ratios remained similar, except for requiring antibiotics which was no longer significant for ≥ ACE categories, Table 5.

The calculated E-value for the adjusted odds ratio for preeclampsia of 4.21 was 7.9.

Missing data were minimal overall. Of the 1,253 women, 23 (1.8%) did not respond to all ACE questions. Of these, 16 had missing data for one ACE category, while the remaining participants had missing data for two to six ACE categories. The ACE categories with the highest proportion of missing data were witnessing domestic violence (8 non-respondents, 0.64%), sexual abuse (7 non-respondents, 0.56%), and household mental illness (6 non-respondents, 0.48%). In a sensitivity analysis missing ACE categories were coded as not experienced, and the results remained identical. The direction and strength of the associations previously noted did not differ in any statistically significant way.

**Table 4. Results of multiple ordinal logistic regression for examining association and effect estimation, among women with exposure of 1-3 and ≥4 Adverse Childhood Experience (ACE) categories, with the group of women exposed to 0 ACE categories as reference.**

| Outcomes | 1-3 ACE categories | | | ≥4 ACE categories | | |
|---|---|---|---|---|---|---|
| | aOR | 95% CI | p | aOR | 95% CI | p |
| Onset of labor | | | | | | |
| Spontaneous | 0.77 | 0.58-1.03 | 0.08 | 0.76 | 0.50-1.16 | 0.219 |
| Induction | 1.14 | 0.82-1.60 | 0.42 | 1.37 | 0.85-2.20 | 0.196 |
| End of labor | | | | | | |
| Vaginal birth, non-instrument-assisted | 0.80 | 0.60-1.06 | 0.123 | 0.72 | 0.48-1.10 | 0.129 |
| Vacuum extraction | 0.88 | 0.56-1.38 | 0.581 | 0.89 | 0.45-1.78 | 0.742 |
| Emergency Caesarean delivery | 1.33 | 0.85-2.09 | 0.208 | *2.02* | 1.13-3.63 | *0.018* |
| Elective Caesarean delivery | 1.46 | 0.93-2.30 | 0.10 | 1.09 | 0.54-2.23 | 0.804 |
| Complications of labor[1] | | | | | | |
| None | 0.85 | 0.65-1.11 | 0.226 | *0.62* | 0.42-0.91 | 0.014 |
| Bleeding >1000 ml | 1.32 | 0.82-2.15 | 0.251 | *2.00* | 1.07-3.75 | *0.030* |
| Vaginal delivery | 1.11 | 0.63-1.95 | 0.725 | 1.61 | 0.74-3.47 | 0.228 |
| Caesarean delivery | 2.14 | 0.74-6.21 | 0.162 | *3.54* | 1.01-12.39 | *0.048* |
| Retained placenta | 0.79 | 0.41-1.50 | 0.467 | 1.70 | 0.78-3.61 | 0.182 |
| Placental abruption | *) | *) | | 3.64[**] | 0.66-20.17 | 0.139 |
| Preeclampsia | 1.55 | 0.69-3.44 | 0.286 | *4.21* | 1.73-10.25 | 0.002 |
| Obstetric anal sphincter injury[2] | 0.84 | 0.47-1.52 | 0.565 | 0.57 | 0.19-1.67 | 0.304 |
| Requiring antibiotics | *2.62* | 1.21-5.68 | *0.014* | *3.14* | 1.19-8.32 | *0.021* |
| Neonatal respiratory support | 1.70 | 0.94-3.06 | 0.080 | 1.59 | 0.68-3.75 | 0.286 |
| Infant admitted to the NICU[3] | 1.42 | 0.84-2.39 | 0.185 | 1.53 | 0.73-3.20 | 0.256 |
| Other | 1.23 | 0.74-1.95 | 0.389 | 1.42 | 0.74-2.74 | 0.290 |

[1] In connection to delivery.

[2] In connection to vaginal delivery.

[3] Neonatal intensive care unit.

*) Odds ratio could not be calculated.

**) Odds ratio with groups of ACE categories 1–3 and ≥4 together.

Results are presented as adjusted odds ratios (aOR) with 95% confidence intervals (CI). All aOR's are adjusted for ethnic origin, as the minimal sufficient adjustment set for estimating the total causal effect.

## Discussion

Exposure to four or more ACE categories was associated with a higher incidence of adverse perinatal outcomes. Women with four or more ACEs had 2.02 times higher odds of emergency Caesarean delivery, 2.00 times higher odds of experiencing abnormal postpartum hemorrhage and 4.21 times higher odds of preeclampsia compared to women with zero ACEs. These findings suggest that the increased adverse perinatal outcomes among women with multiple ACEs may reflect a potential negative impact of ACEs on placental development.

The higher rate of emergency Caesarean deliveries may partially be attributed to the increased incidence of perinatal complications. Additionally, fear of labor could contribute to the likelihood of Caesarean delivery [22]. It is also plausible that mothers with multiple ACEs face challenges in advocating for themselves during childbirth.

There are several potential mechanisms through which ACEs may influence pregnancy outcomes. First, ACEs increase the likelihood of negative health behaviors during pregnancy, which are associated with a higher risk of complications [13,17]. Second, childhood adversity has been linked to alterations in epigenetics and both neuroendocrine and immune

**Table 5. Results of a sensitivity analysis by multiple ordinal logistic regression for examining association and effect estimation, among women with exposure of 1-3 and ≥4 Adverse Childhood Experience (ACE) categories, with the group of women exposed to 0 ACE categories as reference.**

| Outcomes | 1-3 ACE categories | | | ≥4 ACE categories | | |
|---|---|---|---|---|---|---|
| | aOR | 95% CI | p | aOR | 95% CI | p |
| Onset of labor | | | | | | |
| Spontaneous | 0.75 | 0.55-1.04 | 0.062 | 0.80 | 0.50-1.27 | 0.339 |
| Induction | 1.12 | 0.79-1.59 | 0.528 | 1.17 | 0.69-1.97 | 0.563 |
| End of labor | | | | | | |
| Vaginal birth, non-instrument-assisted | *0.73* | 0.54-0.98 | *0.039* | 0.69 | 0.44-1.09 | 0.110 |
| Vacuum extraction | 0.89 | 0.55-1.42 | 0.613 | 0.90 | 0.42-1.91 | 0.782 |
| Emergency Caesarean delivery | 1.51 | 0.93-2.44 | 0.093 | *2.00* | 1.03-3.85 | *0.039* |
| Elective Caesarean delivery | *1.68* | 1.02-2.77 | *0.040* | 1.34 | 0.61-2.94 | 0.461 |
| Complications of labor[1] | | | | | | |
| No | 0.86 | 0.65-1.14 | 0.285 | *0.61* | 0.40-0.94 | *0.024* |
| Bleeding >1,000 ml | 1.44 | 0.87-2.39 | 0.160 | *2.20* | 1.12-4.33 | *0.023* |
| Vaginal delivery | 1.12 | 0.63-2.00 | 0.689 | 1.72 | 0.77-3.82 | 0.186 |
| Caesarean delivery | 3.53 | 0.92-13.57 | 0.066 | *4.86* | 1.02-23.11 | *0.047* |
| Retained placenta | 0.69 | 0.35-1.34 | 0.276 | 1.75 | 0.78-3.93 | 0.177 |
| Placental abruption | *) | *) | | 4.78[**] | 0.77-29.82 | 0.094 |
| Preeclampsia | 1.51 | 0.66-3.43 | 0.324 | *4.70* | 1.81-12.18 | *0.002* |
| Obstetric anal sphincter injury[2] | 0.83 | 0.44-1.54 | 0.561 | 0.68 | 0.22-2.06 | 0.494 |
| Requiring antibiotics | *2.27* | 1.02-5.04 | *0.045* | 1.68 | 0.53-5.35 | 0.381 |
| Neonatal respiratory support | 1.55 | 0.83-2.88 | 0.168 | 1.02 | 0.36-2.88 | 0.970 |
| Infant admitted to the NICU[3] | 1.34 | 0.78-2.30 | 0.282 | 1.56 | 0.71-3.44 | 0.272 |
| Other | 1.24 | 0.76-2.02 | 0.389 | 1.43 | 0.71-2.87 | 0.320 |

[1] In connection to delivery.

[2] Caesarean deliveries excluded.

[3] Neonatal intensive care unit.

*) Odds ratio could not be calculated.

**) Odds ratio with groups of ACE categories 1–3 and ≥4 together.

Results are presented as adjusted odds ratios (aOR) with 95% confidence intervals (CI). All aOR's are adjusted for age, high education level, ethnic origin, smoking habits, alcohol habits, BMI and number of previous pregnancies.

pathways that may impact pregnancy [23]. For example, a 2015 study by Moog *et al.* identified a connection between childhood trauma and placental-fetal stress physiology, as evidenced by increased production of placental corticotropin-releasing hormone [24]. Another study 2019 by Jones *et al.* found that ACE exposure was associated with accelerated placental aging [25]. Both abnormal placentation and dysfunction of the placenta have been linked to preeclampsia [26,27].

The association between ACEs and preeclampsia aligns with findings from previous studies 2021 by Miller *et al.* [15] and 2023 by Swedo *et al.* [17], which reported links between ACEs and hypertensive disorders of pregnancy. Additionally, one study indicated an increased risk of maternal perinatal complications among women with six or more ACE categories [28].

Research on the relationship between ACEs, birth complications, and mode of delivery remains limited. A 2009 study by Heimstad *et al.* [29] identified a strong connection between childhood physical or sexual abuse and mode of delivery, with those reporting abuse being less likely to experience an uncomplicated vaginal delivery. Similarly, a study of 164 mothers found a higher risk of maternal perinatal complications associated with high ACE exposure [28].

This study has several important limitations when interpreting the results. Multiple selection steps in forming the study sample may have introduced bias, as participants might represent a more privileged group compared to those initially approached. Additionally, as with all retrospective studies, there is an inherent risk of recall bias. These limitations may have led to an underestimation of the true incidence of ACE exposure and associated outcomes. However, we believe that the limitations did not compromise the validity of the reported associations.

Since perinatal outcomes were reported one year postpartum, there is a possibility that some complications were forgotten, while certain stressful memories have been exaggerated. However, the recall bias for perinatal outcomes is expected to be consistent across all participants. Reporting birth complications and experiences one year postpartum may also have an advantage – the complications reported are likely those that had a lasting impact on the mothers and are therefore more relevant.

A longitudinal study comparing prospective and retrospective ACE measures found modest agreement in reporting. Nonetheless, both measures were shown to be associated with adult health outcomes in prior studies. Retrospective ACE measure were more strongly predictive of subjective, self-reported outcomes, while prospective ACE measures better predicted objective outcomes, irrespective of whether the adversity was recalled [30].

This study lacks data on the mothers' socioeconomic status during childhood, which prevented adjustment for this factor. Additionally, other unmeasured confounders, such as the mothers' parents' experiences of ACEs, may have introduced bias to the results. The definition of ACEs in this study does not encompass all types of childhood adversity but is based on the questions commonly used in prior research. While ACEs were reported retrospectively, measures such as emergency Caesarean delivery remain objective.

To address potential confounding, two sensitivity analyses was conducted. First, for preeclampsia, the observed odds ratio of 4.21 could theoretically be explained by a confounder with a strong association with both the exposure and outcome, featuring an odds ratio of 7.9 times greater than the adjusted odds ratio. However, weaker confounding associations would not be sufficient to explain this. Therefore, relatively strong confounding relationships would be necessary to fully explain the observed exposure-outcome association with an odds ratio of 4.21. Second, an alternative causality model based on other assumptions about the complexity among potential confounders gave similar adjusted odds ratios as the regression with a minimal set of confounders.

A notable strength of this study is its relatively large and diverse sample, encompassing women of various ethnic origins, education levels, and income brackets. Another strength lies in its broader focus – not only specific adverse events but also on the cumulative impact of exposure to multiple ACEs. Given the tendency of ACEs to co-occur, it is highly relevant to examine a wide range of childhood adversities, beyond individual instances of abuse [31].

This study further contributes to the growing evidence of the detrimental effects of adverse childhood experiences across the individual's lifespan. It underscores the critical need to prevent ACEs and adopt a lifetime perspective on health, including offering trauma treatment options.

Pregnancy presents a valuable opportunity to identify women who have experienced childhood adversities. In Sweden's prenatal care program, questions are already included about current experiences of sexual, physical or mental abuse. Our study highlights the importance not only of these questions but also emphasizes the need to expand then to include experiences of childhood abuse, neglect, and household dysfunction. The low rate of missing responses to ACEs-related questions in this study suggests that patients are willing to answer such questions. Further, research indicates that both patients and clinicians view screening for ACEs positively. In a study by Flanagan *et al*. [32], most patients were comfortable answering ACE-related questions. Clinicians also emphasized that effective screening requires simultaneous assessment of resilience factors and access to appropriate resources, such as mental health support, parental guidance, and social services [32].

Future research should focus on exploring resilience factors and strategies to reduce the negative effects of ACEs. Additionally, it should also examine how the healthcare system can better look to support mothers-to-be affected by childhood adversity.

## Conclusions

Exposure to multiple adverse childhood experiences was associated with increased rates of adverse perinatal outcomes, including higher risks for emergency Cesarean sections, requiring antibiotics, and labor complications. Maternal health services have the opportunity to identify these individuals and offer extra care to reduce their risks.

## Key message

Maternal adverse childhood experiences (ACEs) are strongly associated with negative perinatal outcomes. Women with multiple ACEs exhibit higher odds of postpartum hemorrhage in connection with Caesarean delivery, requiring antibiotics, and, most notably, preeclampsia, suggesting that ACEs may negatively influence placental development.

## Supporting information

**S1 File. Minimal data set.**
(XLSX)

## Acknowledgments

Proof reading and copy-editing: Sarah Evander BSc (Hons), Imperial College, London, UK.

## Author contributions

**Conceptualization:** Amanda Troëng, Jessica Dolk, Per Kristiansson.

**Data curation:** Amanda Troëng, Per Kristiansson.

**Formal analysis:** Amanda Troëng, Marie-Therese Vinnars, Johan Hallqvist, Per Kristiansson.

**Methodology:** Jessica Dolk, Marie-Therese Vinnars, Per Kristiansson.

**Project administration:** Per Kristiansson.

**Supervision:** Johan Hallqvist.

**Writing – original draft:** Amanda Troëng.

**Writing – review & editing:** Jessica Dolk, Marie-Therese Vinnars, Johan Hallqvist, Per Kristiansson.

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
