## [Decision Letter · Decision Letter 0]

5 Mar 2025

PLOS ONE

Dear Dr. Kristiansson,

Thank you for submitting your manuscript to PLOS ONE. After careful consideration, we feel that it has merit but does not fully meet PLOS ONE’s publication criteria as it currently stands. Therefore, we invite you to submit a revised version of the manuscript that addresses the points raised during the review process.

This is an interesting manuscript, which has two well-written reviews that suggest that major revisions are needed before it can be considered ready for publication. I agree with the reviewers. Whilst all the comments from the reviewers need addressing in a revised version,  I believe that of particular importance is the comment from reviewer 1 that "the main analyses appear to lack critical covariates, which could impact birth-outcomes directly, and their exclusion is not sufficiently justified" and from reviewer 2 that "...the authors controlled only for origin in the adjusted logistic regressions, however, the regressions could also be adjusted for age, education, income, and relationship status, as these are common potential confounders." These comments need to be dealt with particularly robustly in the revised version of the manuscript, as the final decision as to whether or not to accept the manuscript for publication will be weighted towards how they are addressed. I would also suggest that the manuscript needs careful editing relating to grammar and use of language. 

We look forward to receiving your revised manuscript.

Kind regards,

Clive J. Petry, PhD

Academic Editor

PLOS ONE

2. Please match your authorship list in your manuscript file and in the system.

4. Thank you for uploading your study's underlying data set. Unfortunately, the repository you have noted in your Data Availability statement does not qualify as an acceptable data repository according to PLOS's standards.

Reviewers' comments:

Reviewer's Responses to Questions

**Comments to the Author**

1. Is the manuscript technically sound, and do the data support the conclusions?

Reviewer #1: Partly

Reviewer #2: Yes

2. Has the statistical analysis been performed appropriately and rigorously?

Reviewer #1: No

Reviewer #2: Yes

3. Have the authors made all data underlying the findings in their manuscript fully available?

Reviewer #1: Yes

Reviewer #2: Yes

4. Is the manuscript presented in an intelligible fashion and written in standard English?

Reviewer #1: No

Reviewer #2: Yes

Reviewer #1: Overall, this study offers additional insight into the specific effects of the early-life adversity on childbirth-outcomes. While these findings appear novel, relevant, and important, there are many major revisions required for the clarity and quality of analytical approaches used. In particular, the main analyses appear to lack critical covariates, which could impact birth-outcomes directly, and their exclusion is not sufficiently justified. The manuscript would greatly benefit from additional transparency of statistics report and clarity of writing throughout the draft. More detailed comments can be found below:

1. There appears to the an error with the attrition reported? “Those who answered Q1 (3389) received Q2. The women who answered Q2 received Q3 (2018/3390) whereof 1253 women (1253/2018) answered Q3”

2. Could you please provide statistics/values for Q1 vs Q3 group differences? “Women answering Q3 who were hence included in the study were different as a group compared to women answering only Q1. The included women were slightly older with higher attained 8 education level and household income, and they were more often of European origin. Their alcohol and cigarette habits were lower than the women who did not answer all questionnaires. The body mass index (BMI) and the proportion women in a relationship were similar.”

3. The analyses do not control for previous pregnancies/complications? Either needs to be empirically explored or listed as a limitation

4. “Other unmeasured confounders may also have biased our results” — Can you provide some examples or speculation?

5. “To analyze the odds for several outcomes among mothers exposed to 1-3 ACE categories or ≥4 ACE categories with women exposed to 0 ACE categories as reference we used ordinal logistic regression analyses reported as crude (OR) and adjusted odds ratios (aOR) and their corresponding 95% confidence intervals (CI). The potential confounding factor adjusted for was Non-Swedish origin. A p-value of <0.05 was considered significant.”

Table 2 shows several group differences in variables that could impact birth outcomes, but these are not controlled for in the ordinal logistic regression model. Why was only Non-swedish origin accounted for? Sensitivity analyses should be conducted to see if the odds ratios remain significant after controlling for these factors (such as smoking, alcohol use, and BMI) OR it needs to be clearly explained why these were not controlled for. These are described as covariates, but then do not seem to be included in the ordinal logistic regression analyses reported as crude (OR) and adjusted odds ratios (aOR) analyses.

6. The Directed acyclic graph analysis is also very unclear. This appears to be used to justify why the only co-variate included was non-Swedish origin, but this rationale is not overtly clear and there are no statistics to back this claim.

7. Finally, the manuscript requires extensive copy editing that is beyond the scope of what I can point out as a reviewer. In many instances the syntax of sentences interferes with the ability to clearly comprehend the ideas attempted to be communicated.

Reviewer #2: This manuscript describes a retrospective cohort study which investigated the impact of adverse childhood experiences on mode of delivery and delivery complications among a sample of postpartum women in Sweden. While the findings make an important contribution to the literature, there are several points of feedback which require attention by the authors and should be addressed to improve the accuracy, clarity, and cohesiveness of the paper. Please see points of feedback below.

• The sentence in the introduction, “Exposure to ACE is linked to adverse health outcomes, higher risk of several common public health diseases as well as risk behaviors and lower socioeconomic status” is vague and should be revised to specify the adverse health outcomes, public health diseases, and risk behaviors of relevance, and citations should be provided.

• The authors switch back and forth between the acronym “ACE” and “ACEs.” This should be made consistent throughout the paper. While the acronym “ACE” was used in the original ACE study (Felitti et al., 1998), “ACEs” is the commonly utilized acronym in the literature. I recommend that the authors change the acronym from “ACE” to “ACEs” throughout the manuscript for consistency purposes.

• The following sentence in the abstract, “Studies on adverse childhood experiences and perinatal outcomes are scarce” as well as the following sentence in the introduction, “However, there is still limited research about the effect of multiple adverse childhood experiences on perinatal outcomes” should be revised and clarified because these sentences are not fully accurate. While I agree with the authors that there is more limited literature specifically on the effects of ACEs on mode of delivery or delivery complications, there is quite a bit of literature on other perinatal outcomes, especially adverse birth outcomes (see Sulaiman et al., 2021, in addition to Christaens et al., 2015; Swedo et al., 2022; Smith et al., 2016; and Christiaens et al., 2015 which were cited in the manuscript references). The authors should clarify and revise these sentences in acknowledgement of the previous literature.

• The description of the derivation of the analytic sample and the participation/response rates written in the materials and methods section is unclear. There are multiple denominators referenced (5796, 3389, etc) and several numerators (i.e., 1253 and 2018)- it would be helpful to create an analytic flow chart to display this information so it is clear what the source population is and how the study sample is derived (and also to clarify how the sample size was reached for each questionnaire).

• The results of the assessment of bias written in the paragraph on page 8 (cited below) should include significance testing (i.e., p values) or should be included in a supplementary appendix. For grammatical accuracy, the word "of" can be inserted into the following sentence: “Women answering Q3 who were hence included in the study were different as a group compared to women answering only Q1. The included women were slightly older with higher attained education level and household income, and they were more often of European origin. Their alcohol and cigarette habits were lower than the women who did not answer all questionnaires. The body mass index (BMI) and the proportion [of] women in a relationship were similar.”

• On page 8 when the authors state, “To create a Swedish version, the questions were translated from English to Swedish and then translated back to English. This tested the accuracy by comparison with the original questionnaire,” it would be helpful to have some additional details here about the authors’ validation efforts. The process as described sounds like forward translation and back-translation processes were conducted, which can be specified techniques in this section. The authors should elaborate who conducted these processes, and were bilingual experts consulted?

• The following sentence on page 8 is not clear: “Each ACE category was coded as not experienced until any of the criteria described in detail elsewhere (3) was true.” What do the authors mean when they state, “…until any of the criteria described in details elsewhere was true?”

• Under the category of “Obstetric complications” what data is considered “other”? This was not clear in the methods or results section.

• Please define “SEK.” All acronyms should be defined prior to subsequent use: “Monthly household income was presented on an 11-point Likert scale between 0 SEK and >100

000 SEK with 10 000 SEK increments as the total monthly household income before taxes” (p. 10).

• For all exposure variables, outcome variables, and covariates, it would be helpful to include in an appendix table how the variables were operationalized and how the question was asked to capture the variable.

• On page 11 the authors state: “Additionally, this avoids introducing bias by adjusting for mediators.” However, in the following sentence, the authors state, “As an example, BMI was assessed as a mediator and hence not adjusted for.” These sentences appear contradictory. It is not recommended to adjust for mediators as this can introduce bias, since mediators are on the causal pathway between the exposure and the outcome. Please elaborate.

• The ethics statement is unclear as written. The authors state that the study received approval, but that “return of the completed questionnaires was regarded as informed consent.” What was the process used to obtain informed consent and how was informed consent documented?

• The results section is too brief and not comprehensive. For example, in the results section for Table 4, the authors should acknowledge the findings across all the outcomes (regardless if they were significant or not). The description of the results section does not include statistical figures or the measures of association depicted in the tables.

• The primary measure of association for the outcomes in Table 4 are odds ratios, yet there is no interpretation of the odds ratios in the results or discussion. This is a significant limitation that needs to be addressed. In the discussion, the authors use the term “risks” and then “rates” but odds are not risks or rates. For example, the interpretation for preeclampsia should be written as follows, “Women who reported 4 or more ACEs had 4.21x higher odds of preeclampsia compared to women who reported 0 ACEs.”

• The authors controlled only for origin in the adjusted logistic regressions, however, the regressions could also be adjusted for age, education, income, and relationship status, as these are common potential confounders.

• Since the impacts on placental development were not directly assessed, the following sentence should be revised to be more cautious: “The increased risk of abnormal postpartum hemorrhage and the substantial risk of preeclampsia among women with multiple ACEs indicates a negative impact of reported ACE categories on placental development [suggested revision: "The increased risk of abnormal postpartum hemorrhage and the substantial risk of preeclampsia among women with multiple ACEs suggests that ACEs may have adverse impacts on placental development"].” Please cite relevant references that support this explanation (such as the references that are mentioned in the following paragraphs about placental dysregulation).

The following sentences should be revised for grammatical accuracy:

• “During the recruitment period, between September 2012 and July 2013, 5796 women were enrolled of which 3389 (62%) chose to participate and did not meet the exclusion criteria [of] not speaking Swedish, miscarriage[,] or multiple birth[s].”

• The following sentence is unclear. Did women answer the first questionnaire at 9.6 weeks on average, the second questionnaire at 31.9 weeks of gestation, and the third at 52.8 weeks postpartum? If this is what the authors meant, it should be clarified to say so.

“The midwives at the antenatal clinics distributed the first questionnaire (Q1) to the women. The second questionnaire (Q2) and the third questionnaire (Q3) were sent by post together with a prepaid envelope to women who had answered the previous questionnaire. Non-responders were sent a reminder within two weeks. Those who answered Q1 (3389) received Q2. The women who answered Q2 received Q3 (2018/3390) whereof 1253 women (1253/2018) answered Q3. The 1253 women answered all three questionnaires at in average 9.6 and 31.9 weeks of gestation and 52.8 weeks postpartum.”

• The following sentence could be revised to include the word “status” to clarify the variable pertaining to relationship status: “The third questionnaire (Q3) requested information about adverse childhood experiences, recent pregnancy and birth, date of birth, weight, alcohol use, smoking, relationship [status] and household income.

**Do you want your identity to be public for this peer review?** For information about this choice, including consent withdrawal, please see our Privacy Policy

Reviewer #1: No

Reviewer #2: No

---

## [Author Response · Author response to Decision Letter 1]

29 May 2025

Re: PONE-D-24-51497

Dear Editor,

Thank you and the reviewers for your encouraging feedback!

Please find our responses to the reviewers’ comments below. The changes are marked with yellow underline.

Reviewer #1

1. An error with the attrition reported.

Answer: On page 6, 2nd paragraph, the figure regarding “Q2 received Q3” is changed to “3389”.

2. Statistics for group differences for Q1 and Q3.

Answer: On page 7, 1st paragraph, statistics highlighting group differences between women responding to Q3 and those answering Q1 only have been included.

3. Control for previous pregnancies/complications? Empirically explored or listed as a limitation.

Answer: We reference the response given under item 5.

4. “Other unmeasured confounders may also have biased our results” Provide examples.

Answer: On page 28, in the 2nd paragraph, we have added ”as the mothers’ parents’ experiences of ACEs”.

5. Selection of confounders? DAG? Why not control for smoking, alcohol use, and BMI?

Answer: On page 11, 1st paragraph, the following text is added:

“The selection of confounders followed established principles, such as pretreatment criterion and causal diagram. Applying the pretreatment criterion, parental origin and socioeconomic status were identified as variables that precede exposure to adverse childhood experiences (ACEs). Using a causal diagram and the “back-door path criterion,” these two factors were further confirmed as key for mitigating confounding bias. The diagram also suggested that variables such as a new partner, stress during pregnancy, alcohol consumption, smoking habits, BMI, education level, age, parity, and pre-pregnancy hypertension act as mediators. Adjusting for these mediators could introduce bias. Given the absence of data on parental socioeconomic status, origin was selected as the confounder for adjustment in the regression analyses.

A sensitivity analysis was conducted that included an alternative causality model based on other assumptions about the complexity among potential confounders as part of a chain of confounders or mediators. In this model, adjustments were made for age, high educational level, smoking habits, alcohol habits, body mass index and number of previous pregnancies in addition to adjustments in the causal diagram model.”

On page 11, 3rd paragraph we have added information of an additional sensitivity analysis:

“In addition, a sensitivity analysis was conducted to evaluate the robustness of the association to potential unmeasured and uncontrolled confounding, using the E-value calculation, by the formula E-value=OR + sqrt (OR x (OR-1)).”

On page 20, 3rd paragraph we have added “The calculated E-value for the adjusted odds ratio for preeclampsia of 4.21 was 7.9.”

On page 28, 3rd paragraph we have added the following text:

“To address potential confounding, a sensitivity analysis was conducted. For preeclampsia, the observed odds ratio of 4.21 could theoretically be explained by a confounder with a stron association with both the exposure and outcome, featuring an odds ratio of 7.9 times greater than the adjusted odds ratio. However, weaker confounding associations would not suffice. Therefore, relatively strong confounding relationships would be necessary to fully explain the observed exposure-outcome association with an odds ratio of 4.21. Second, an alternative causality model based on other assumptions about the complexity among potential confounders gave similar adjusted odds ratios as the regression with a minimal set of confounders.”

6. DAG is unclear

Answer: Referencing the response provided under item 5 clarifies the DAG.

7. Required copy editing

Answer: The manuscript has undergone a thorough linguistic review. Hopefully, this has enhanced the reader's comprehension.

Reviewer #2

1. Introduction, ACE outcomes to be specified.

Answer: On page 5, 1st paragraph references are added to the specific adverse health outcomes.

2. The acronym ACEs would be used.

Answer: Changed to ACEs throughout where appropriate.

3. Abstract. Clarify and revise these sentences in acknowledgement of the previous literature.

Answer: In the Abstract Introduction: “scarce” is changed to “remains limited, and the effect of cumulative ACEs on perinatal outcomes is unknown”.

In addition, on page 5, 3rd paragraph in the Introduction we have changed the wording to: “However, research on the impact of multiple ACEs on perinatal outcomes remains limited.”

4. Attrition reported.

Answer: We refer to the response provided to Reviewer #1 under item 1. Adjusting the denominators simplifies tracking attrition, thereby reducing the need for a flowchart.

5. The results of the differences among women answering Q3 and Q1 only, should include significance testing.

Answer: On page 7, 1st paragraph, the actual proportions and p-values have been incorporated.

6. Who conducted translation process, bilingual?

Answer: On page 7, 2nd paragraph the following text is added:

“This process involved three translators: one native English speaker with Swedish as a second language, and two native Swedish speakers with English as their second language. The translated questionnaire was then compared to the original to ensure accuracy.”

7. What do the authors mean with “until any of the criteria described in details elsewhere was true.

Answer: On page 8, 1st paragraph this sentence is added: “Each ACE category was initially coded as not experienced unless any of the criteria, described in an earlier study by Andersson et al in 2021 (3) were met”.

8. Obstetric complications. What data is considered in “other”

Answer: On page 8, 2nd paragraph the following text is added:

“The option “other” provided no further details and was selected by 93 of 1,235 women (7.5%).”

9. Please, define “SEK”

Answer: On page 10, 2nd paragraph: “Swedish krona” is added.

10. An appendix table for all variables.

Answer: In the Methods section from page 7, 3rd paragraph to page 10, 4th paragraph, we have included an explanation of how the various questions were formulated and operationalized.

11. Avoid introducing bias.

Answer: We reference the response provided to Reviewer #1 under item 5.

12. The ethics statement. What was the process used to obtain informed consent and documented?

Answer: On page 13, 1st paragraph, the process for obtaining informed consent has been thoroughly detailed.

13. The results section is too brief and not comprehensive. Table 4

Answer: The results are enhanced with statistical figures where relevant and include an additional sentence referring to Table 4.

14. There is no interpretation of the OR in the discussion. Avoid “risk” and “rates”.

Answer: On page 26, the wording of the 1st paragraph has been revised according to your suggestion.

15. Control also for age, education, income and relationship status, as common potential confounders.

Answer: We reference the response provided to Reviewer #1 under item 5.

16. Since the impacts on placental development were not assessed, revise the sentence.

Answer: On page 26, 1st paragraph, the word “indicates” is changed to “suggest”.

17. Revise sentences:

Answer: On page 6, 1st paragraph, the sentence is changed according to the suggestion from the reviewer.

18. Unclear sentence

Answer: On page 6, 2nd paragraph, the sentence has been revised for clarity and comprehensibility based on reviewers’ suggestions

19. Miss of a word.

Answer: On page 6, 3rd paragraph, the word “status” is added to the sentence.

On behalf of the entire group of authors

Kind regards,

Per Kristiansson

Corresponding author

---

## [Decision Letter · Decision Letter 1]

15 Sep 2025

Maternal adverse childhood experiences and perinatal outcomes: A retrospective inception cohort study.

PONE-D-24-51497R1

Dear Professor Kristiansson,

We’re pleased to inform you that your manuscript has been judged scientifically suitable for publication and will be formally accepted for publication once it meets all outstanding technical requirements.

Kind regards,

Adewale Olufemi Ashimi, MBBS, MPH, PhD, FWACS

Academic Editor

PLOS ONE

Additional Editor Comments (optional): I declare no conflict of interest.

Reviewer #1:

Reviewer #3:

Reviewer #4:

Reviewers' comments:

Reviewer's Responses to Questions

**Comments to the Author**

Reviewer #1: All comments have been addressed

Reviewer #3: All comments have been addressed

Reviewer #4: All comments have been addressed

2. Is the manuscript technically sound, and do the data support the conclusions?

Reviewer #1: Yes

Reviewer #3: Yes

Reviewer #4: Yes

3. Has the statistical analysis been performed appropriately and rigorously?

Reviewer #1: Yes

Reviewer #3: Yes

Reviewer #4: Yes

4. Have the authors made all data underlying the findings in their manuscript fully available?

Reviewer #1: Yes

Reviewer #3: No

Reviewer #4: Yes

5. Is the manuscript presented in an intelligible fashion and written in standard English?

Reviewer #1: Yes

Reviewer #3: Yes

Reviewer #4: Yes

Reviewer #1: Authors have addressed all reviewer comments. The manuscript is much clearer and statistically sound. Analyses and conclusions are much easier to follow.

Reviewer #3: Page 5, paragraph 5 (line 2), states, Recently, growing attention has been pain to the adverse effects of ACEs on pregnancy and perinatal outcomes. The highlighted word doesn’t seem to be appropriate.

How did the authors resolve the issue of recall of ACEs by the respondents?

Reviewer #4: in my opinion, all the issues that were raised by the reviewers have been adequately addressed in this revised manuscript.

**Do you want your identity to be public for this peer review?** For information about this choice, including consent withdrawal, please see our Privacy Policy

Reviewer #1: No

Reviewer #3: **Yes: ** Dr. Emmanuel Ajuluchukwu Ugwa

Reviewer #4: No

---

## [Editor Report · Acceptance letter]

PONE-D-24-51497R1

PLOS ONE

Dear Dr. Kristiansson,

I'm pleased to inform you that your manuscript has been deemed suitable for publication in PLOS ONE. Congratulations! Your manuscript is now being handed over to our production team.

Kind regards,

on behalf of

Dr. Adewale Olufemi Ashimi

Academic Editor

PLOS ONE